# How to Identify Advanced Fibrosis in Adult Patients with Non-Alcoholic Fatty Liver Disease (NAFLD) and Non-Alcoholic Steatohepatitis (NASH) Using Ultrasound Elastography—A Review of the Literature and Proposed Multistep Approach

**DOI:** 10.3390/diagnostics13040788

**Published:** 2023-02-19

**Authors:** Madalina-Gabriela Taru, Lidia Neamti, Vlad Taru, Lucia Maria Procopciuc, Bogdan Procopet, Monica Lupsor-Platon

**Affiliations:** 1Hepatology Department, Regional Institute of Gastroenterology and Hepatology “Octavian Fodor”, 400162 Cluj-Napoca, Romania; 2Faculty of Medicine, “Iuliu Hatieganu” University of Medicine and Pharmacy, 400012 Cluj-Napoca, Romania; 3Department of Medical and Surgical Sciences, University of Bologna, 40126 Bologna, Italy; 4Division of Gastroenterology and Hepatology, Department of Medicine III, Medical University of Vienna, 1090 Vienna, Austria; 5Christian Doppler Lab for Portal Hypertension and Liver Fibrosis, Medical University of Vienna, 1090 Vienna, Austria; 6Medical Imaging Department, Regional Institute of Gastroenterology and Hepatology “Octavian Fodor”, 400162 Cluj-Napoca, Romania

**Keywords:** non-alcoholic fatty liver disease, non-alcoholic steatohepatitis, ultrasound elastography, vibration-controlled transient elastography, point shear wave elastography, two-dimensional shear wave elastography, fibrosis, steatosis, inflammation

## Abstract

Non-alcoholic fatty liver disease (NAFLD), and its progressive form, non-alcoholic steatohepatitis (NASH), represent, nowadays, real challenges for the healthcare system. Liver fibrosis is the most important prognostic factor for NAFLD, and advanced fibrosis is associated with higher liver-related mortality rates. Therefore, the key issues in NAFLD are the differentiation of NASH from simple steatosis and identification of advanced hepatic fibrosis. We critically reviewed the ultrasound (US) elastography techniques for the quantitative characterization of fibrosis, steatosis, and inflammation in NAFLD and NASH, with a specific focus on how to differentiate advanced fibrosis in adult patients. Vibration-controlled transient elastography (VCTE) is still the most utilized and validated elastography method for liver fibrosis assessment. The recently developed point shear wave elastography (pSWE) and two-dimensional shear wave elastography (2D-SWE) techniques that use multiparametric approaches could bring essential improvements to diagnosis and risk stratification.

## 1. Introduction

Non-alcoholic fatty liver disease (NAFLD) is causing a substantial burden worldwide. As shown in a recently published meta-analysis, the prevalence of NAFLD has increased over time, from 25.5% in 2005 to an alarming 37.8% in 2016 or later [1], presenting the same rising trend as reported before, from 20.1% to 26.8% between 2000 and 2015 [2] and from 21.9% to 37.3% between 1991 and 2019 [3], respectively. 

NAFLD progresses to non-alcoholic steatohepatitis (NASH) in 25% of cases within 3 years, and NASH already accounts for 12% of all liver transplantations in Europe [4]. Contrary to hepatitis C and other liver-specific disorders, where a single etiological factor is related to the liver damage, NASH is a multifactorial, complex metabolic entity that forms part of a systemic disease, thus raising challenges for disease management and monitoring [5]. In the history and progression of the disease, NASH connotes a liver injury, which can lead to clinically significant portal hypertension (CSPH), cirrhosis, hepatocellular carcinoma, and further decompensation [6]. 

Patients with compensated NASH-related cirrhosis stay in this state for a mean time of 4 years and have a 10% per year risk of progression to decompensation or death. Those experiencing the first decompensation event have a three-fold higher risk of further progression [7]. 

The diagnosis of NAFLD is traditionally based on the histopathological changes in the liver, which is evaluated by liver biopsy (LB) and is defined as triglyceride accumulation in more than 5% of the hepatocytes [8]. The progressive form of NAFLD, NASH, requires, as well as a standard diagnosis, a histological assessment, which is usually performed in the context of suspected fatty liver and is typically characterized by the presence of steatosis, lobular inflammation, and ballooning with or without fibrosis [9]. 

However, LB is indicated with caution, as it holds the potential for adverse effects, sampling errors, inter and intra-observer variability [10]. The procedure also involves interpretation errors due to the inhomogeneous distribution of the fibrosis. Agreement for the fibrosis stage was found in only 47% of patients when two samples were taken from the right and left hepatic lobes of the same NASH patient, with differences in at least one stage found in 41% of the cases and of one stages in 12% of the cases [11]. 

Recently, an international panel of experts has proposed a new nomenclature for NAFLD based on simple and clear diagnostic criteria, namely, the metabolic-associated fatty liver disease (MAFLD) [12]. The definition of MAFLD is based on “positive criteria”, such as the evidence of hepatic steatosis, in addition to one of the following three criteria, namely, overweight/obesity, the presence of type 2 diabetes mellitus, or evidence of metabolic dysregulation. The last criterium is met when at least two features are present alongside: an increased waist circumference, arterial hypertension, hypertriglyceridemia, a low HDL-C, prediabetes, insulin resistance, and subclinical inflammation. Indeed, a need for change in the nomenclature of NAFLD has been suggested in the literature since the early 2000s [13]. MAFLD definition has been considered more practical for identifying patients with fatty liver disease with high risk of disease progression [14,15]. Nevertheless, the definition has not been broadly accepted yet, as a premature change in terminology could lead to ambiguity, discrepancies, and a negative impact on the field [16]. 

As previously reported in most retrospective studies of NAFLD patients, the risk of liver-related events (LREs) significantly increased at fibrosis stage 2 (F2—significant fibrosis) and became exponentially higher when they were transitioning to the advanced fibrosis stages (F3—bridging fibrosis; F4—cirrhosis, respectively) [17,18]. According to other studies, higher mortality rates were registered for F4 (1.76 deaths per 100 person years), and among those at stage F3 (0.89 deaths per 100 person years) [19]. Nevertheless, risk of all-cause and liver-related mortality increased substantially with fibrosis stage, as reported in a recently published meta-analysis [20]. 

Therefore, the fibrosis stage is an important predictor of long-term liver related outcomes and overall mortality. Identifying its advanced stages (≥F3) among patients with NAFLD and NASH is of outmost importance [21]. In this context, novel information provided by non-invasive methods for the evaluation of liver fibrosis may help the clinician in formulating an early and accurate diagnosis, while reducing the number of liver biopsies and proposing a cost-effective surveillance of these patients [22,23,24]. 

In the following pages, we outline the latest developments in the non-invasive assessment of fibrosis, steatosis, and inflammation in adult patients using ultrasound elastography as promising emerging tools for NAFLD and NASH diagnosis. 

## 2. Liver Elastography: The Main Techniques

All liver diseases, focal and diffuse ones, are associated with changes in the structure of the tissue, leading to alterations in the biomechanical properties of the liver, which can be quantified using tissue elastography. 

The elastography techniques can be divided into two main categories, magnetic resonance imaging (MRI)-based elastography techniques and ultrasound (US)-based elastography techniques. 

The proposed elastography MRI-based techniques in liver fibrosis staging are magnetic resonance elastography (MRE) and the magnetic resonance imaging (MRI) index tests such as LiverMultiScanTM (LMS) are used to measure iron-corrected T1 (cT1), diffusion-weighted imaging (DWI), and for the detection of metabolic and liver injury (deMILI). Briefly, MRE generates a quantitative 3D elasticity map that covers the entire liver, it is less dependent on operators, without being affected by air or bones. Despite it presenting a very good performance for diagnosing significant fibrosis (≥F2) (sAUC 0.91, 95% CI 0.80–0.97), advanced fibrosis (≥F3) (sAUC 0.92, 95% CI 0.88–0.95), and cirrhosis (F4) (sAUC 0.90, 95% CI 0.81–0.95) in patients with biopsy-proven NAFLD, when one is considering liver assessments, MRE has a more limited utility compared to that of the ultrasound methods because of their high costs and restricted availability [25]. Therefore, in this review, we focus on the ultrasound-based elastography techniques, as they are the most widely utilized outside the radiology practice. 

By measuring certain intrinsic physical properties of the liver parenchyma, such as stiffness, attenuation, and viscosity, elastography provides a new dimension to the conventional ultrasound examination.

The ultrasound elastography techniques can be divided into quantitative (“shear wave elastography”, SWE) and qualitative (“Strain Elastography” and “Real-time Elastography”) ones [26]. 

The quantitative techniques—vibration-controlled transient elastography (VCTE), 2D shear wave elastography (2D-SWE), and point shear wave elastography (pSWE)—are usually used in the assessment of diffuse liver diseases, while “strain” techniques have a lower applicability in this respect [27]. Due to the efforts of the Quantitative Imaging Biomarkers Alliance (QIBA), the variability of liver stiffness measurements between systems has decreased [28]. 

### 2.1. Vibration-Controlled Transient Elastography (VCTE) 

In the 2000s, vibration-controlled transient elastography (VCTE) was the first tool with which liver stiffness (LS) was measured and studied. It is the only quantitative elastography technique that has not been integrated into a standard ultrasound system. The technique is performed using the Fibroscan^®^ device (Echosens, Paris, France) [29], as shown in Figure 1. It is also the modality with the most available data for evaluating fibrosis in NAFLD and NASH patients, with 53 included studies in a recently published meta-analysis [25]. 

To perform the examination, the patient is placed in a dorsal decubitus position, with the right arm in maximum abduction to best expose the right quadrant. The transducer is placed in direct contact with the skin, perpendicularly to the intercostal space, at a point of maximal hepatic dullness (at least 6 cm thick), free of any large vascular structures (usually the 9–11th intercostal space on the midaxillary line) [30]. There are two types of transducers that are mainly utilized in the adult population, the M (3.5 MHz) and XL (2.5 MHz) probes. The XL probe is usually chosen for patients with a BMI > 30 kg/m^2^, with the machine recommending its utilization [31,32]. 

When one presses the transducer button, a painless mechanical vibration is generated, inducing a train of elastic waves (50 Hz), which propagate through the skin and subcutaneous tissue deep to the liver parenchyma [33]. The shear wave is tracked through multiple acquisitions, requiring 10 valid measurements, with a reported interquartile range (IQR) and interquartile range/median ratio (IQR/M) after 10 conclusive measurements [34]. The time necessary for the train of waves to propagate along the area of interest, as well as the velocity of propagation, are recorded. Based on these recordings, the Young’s modulus (E) is further calculated in kilopascals (kPa), and it clinically corresponds to the LS, with values ranging from 2.5 to 75 kPa [29,35]. An IQR of less than 30% of the median should be obtained for an accurate liver stiffness measurement (LSM), regardless of the success rate (SR), if 10 valid measurements are obtained [26] (Figure 1). 

The equipment can measure the liver stiffness (for the estimation of fibrosis) and the controlled attenuation parameter (CAP) (for the estimation of steatosis). 

The stiffer the tissue is, the higher the values of LS are, while lower values indicate a more elastic liver. An LS of around 4.5–5.5 kPa is reported in the healthy population [26]. 

Using this technique, the LS can be measured for a cylinder of parenchyma of 1 cm diameter and 4 cm height, representing around 1/500 of the entire liver volume [36]. 

The measurements can be performed even by a technician after a training period (approximately 100 cases), but the clinical interpretation of results must always be issued by an expert who considers the demographic data, disease etiology, and biochemical profile at the moment of the examination [37]. 

A necessary condition for a correct assessment is the examination after an overnight fast or at least 3 h of fasting after a meal because a postprandial examination would increase the stiffness value due to increased hepatic blood flow [26,38].

The interobserver concordance was lower in patients with a BMI ≥ 25 kg/m^2^. Using the M probe to measure LSM, lower success rate were obtained among obese subjects [39]. To overcome this drawback, the XL probe was created. The main limiting factors for the XL probe were a skin-to-liver capsule distance of >3.4 cm and extreme obesity (BMI > 40 kg/m^2^) [34,40]. Values obtained with the XL probe were usually lower than those obtained with the M probe, but no recommendation for the cut-off s has been given in the latest published guidelines [26,41]. 

#### 2.1.1. Limitations and Errors of Interpretation 

Liver tissue abnormalities such as edema, inflammation (cytolysis), cholestasis, congestion (heart failure), and infiltrative diseases may interfere with LSM independently of fibrosis, so these factors should be considered when one is interpreting the values of the hepatic rigidity [42,43,44,45,46]. 

The influence of steatosis on LS is still rather controversial; some studies indicate that steatosis may lead to higher LS values [47,48,49], independently of fibrosis, whereas others did not find the same effect [50]. 

#### 2.1.2. Performance of VCTE for Liver Fibrosis Assessment in NAFLD and NASH

In a recently published meta-analysis [25] that evaluated the diagnostic accuracy of five elastography and imaging modalities for non-invasive fibrosis detection in patients with biopsy-proven or suspected NAFLD, the authors reported that the sAUC values for VCTE were 0.82 (95% CI 0.78–0.85, sSe 78%, sSp 72%) for detecting any stage of fibrosis (≥F1), 0.83 (95% CI 0.80–0.87, sSe 80%, sSp 73%) for diagnosing significant fibrosis (≥F2), 0.85 (95% CI 0.83–0.87, sSe 80%, sSp 77%) for diagnosing advanced fibrosis (≥F3), and 0.89 (95% CI 0.84–0.93, sSe 76%, sSp 88%) for diagnosing cirrhosis, respectively. When they were performing a multiple-threshold meta-analysis on the studies that reported more than two cut-off values for VCTE, the authors reported that the sAUC for diagnosing advanced fibrosis (≥F3) was 0.85 (sSe 80%, sSp 75%) and the Youden index was maximized by an 8.7 kPa cut-off. A cut-off of 8.9 kPa was associated with 80% sSe and 77% sSp, and a cut-off of 9.5 kPa was associated with sSe 76% and sSp 80%. These results should be interpreted with some caution as some of the included studies did not use the XL probe or might not have used it according to the manufacturer’s recommendations, and the data presenting recent improvements of the VCTE, such as the continuous CAP algorithm, or its utility in patients with morbid obesity were not included in the study. 

The recently published Baveno VII guidelines [51], which are applicable also to NAFLD and NASH, brought important improvements to LSM by VCTE. LS values < 10 kPa in the absence of other known clinical/imaging signs rule out compensated advanced chronic liver disease (cACLD); values between 10 and 15 kPa are suggestive of cACLD; values >15 kPa are highly suggestive of cACLD. Nevertheless, patients with LS values of 7–10 kPa and with an ongoing liver injury should be monitored on a case-by-case basis for changes indicating the progression to cACLD [51]. The same guidelines described “the rule of five” (10–15–20–25 kPa), which should be used to denote progressively higher relative risks of decompensation and liver-related death, independent of the etiology of chronic liver disease. 

At present, there is a significant heterogeneity among the “in practice” used cut-offs. As recently reported, the cut-offs used for the same non-invasive tests (NITs) for NAFLD risk-stratification varied between clinicians, including those used for VCTE. Of the 28 responders reporting the use of (Fibroscan^®^), 6 (21%) used a single cut-off, and of these, only 3 (50%) reported 8 kPa, with the other outlined variants being 7.2 kPa, 7.8 kPa, and 8.7 kPa. Among the 63% respondents who used lower and upper LS cut-offs, the most common lower cut-off of <8.0 kPa was reported by seven respondents (32%) and the most common higher cut-off of >15 kPa was reported by five respondents (23%) [52]. 

#### 2.1.3. Assessing Steatosis through the Controlled Attenuation Parameter (CAP)

Briefly, the controlled attenuation parameter (CAP) estimates the total ultrasonic attenuation and has been developed as a feature of the FibroScan^®^ device for assessing liver steatosis. CAP is expressed in decibels per meter (dB/m), ranging between 100 and 400 dB/m, with the normal values being under 247 dB/m [53]. Optimal cut-offs were established in a meta-analysis including 2735 patients (of which 20% were diagnosed with NAFLD/NASH) for the prediction of mild (S1), moderate (S2), and severe steatosis (S3). The cut-off values were established at 248 dB/m, (AUC 0.823, Se 68.8%, Sp 82.2%), 268 dB/m, (AUC 0.865, Se 77.3%, Sp 81.2%), and 280 dB/m (AUC 0.882, Se 88.2%, Sp 77.6%) for detecting steatosis ≥S1, ≥S2, and S3, respectively. Steatosis was graded on histology according to the number of affected hepatocytes as: absence of steatosis (S0) (<5 or <10% depending on the trial), S1 (between 5 and 10–33%), S2 (34–66%), and S3 (>66%) [54]. The authors suggested that these cut-offs should be further adapted for particular settings as they were influenced by several covariates. Specifically, it was suggested that practitioners should deduct 10 dB/m from the CAP value for NAFLD/NASH patients, deduct 10 dB/m for diabetes patients, and deduct/add 4.4 dB/m for each unit of BMI above/below 25 kg/m^2^ over the range of 20–30 kg/m^2^. 

### 2.2. Point Shear Wave Elastography (pSWE) 

Among the pSWE techniques, our review will focus on Acoustic Radiation Force Impulse Elastography (ARFI).

This quantitative technique provides a single, unidimensional measurement of tissue elasticity. To perform this examination, the patient is placed in a supine or slight left lateral decubitus position (not more than 30°) with the right arm abducted [27]. The patient is asked to fast for a minimum of 2 h and rest for a minimum of 10 min prior to the examination and to hold their breath in a neutral position during the procedure, as inspiration affects the measurement [26]. The transducer is positioned in an intercostal space, choosing the region of interest (ROI) placement on the B-mode ultrasound image, avoiding vascular structures and at a minimum of 1–2 cm and a maximum of 6 cm beneath the liver capsule [55,56,57,58]. With the ROI selected, pSWE measures the shear wave velocity (SWV) induced by the acoustic radiation propagating in the tissue [59]. The results can be reported in meters per second (m/s) or kilopascals (kPa). The stiffer the tissue is, the higher the share wave velocity is [26]. In general, 10 valid measurements are required, but according to some authors, there was no loss of accuracy when 5 valid measurements were obtained [60]. An IQR/median ratio ≤ 0.3 is considered to be reliable [61]. 

The pSWE results are influenced by the same error factors as any other ultrasound elastography technique is (cytolysis, cholestasis, congestion, or infiltrative liver diseases) [62,63]. Therefore, these factors should be considered when one is interpreting shear wave velocity values. 

There are several types of ultrasound equipment that use a pSWE technique, with the first one being implemented in the Siemens ultrasound systems [64] (Figure 2). As the technique became commercially available over the years, its clinical utility was reported in a substantial number of published studies [65,66,67,68]. A recently developed method for liver stiffness assessment is the pSWE-Elast-PQ (EPQ) technique implemented in the Philips ultrasound systems [69,70,71]. 

The equipment lists the SWV (m/s) in the region of interest, as well as the depth at which the measurement is performed. 

#### Performance of pSWE for Liver Fibrosis Assessment in NAFLD and NASH

The “vendor-neutral rule of four” for the interpretation of ARFI techniques stated that LS ≤ 5 kPa (1.3 m/sec) had a high probability of being normal, LS < 9 kPa (1.7 m/sec), in the absence of other known clinical signs, ruled out cACLD, and values of >13 kPa (2.1 m/sec) were highly suggestive of cACLD; in some patients with NAFLD, the cut-off values for cACLD might have been lower, and a follow-up or additional testing of those with values between 7 and 9 kPa would be recommended. A value of >17 kPa (2.4 m/sec) was suggestive for clinically significant portal hypertension (CSPH). Most studies that used pSWE suggested that a liver stiffness value of less than 7 kPa (1.5 m/sec) could help to rule out significant fibrosis [28]. However, comparing to VCTE, the LS values provided by pSWE have a narrow range (0.5–4.4 m/s), so discriminating certain fibrosis stages is more restricted, thus it is harder to make management decisions. 

The performance of pSWE-Elast-PQ (EPQ) was evaluated in a cohort of biopsy-proven NAFLD patients [72], with reported AUC values of 0.8 (Se 65.5%, Sp 81.3%), 0.72 (Se 75.6%, Sp 61%), 0.69 (Se 75.8%, Sp 47.2%), and 0.79 (Se 75%, Sp 92.7%), respectively, for diagnosing any stage of fibrosis (≥F1), significant fibrosis (≥F2), advanced fibrosis (≥F3), and cirrhosis (F4), with the proposed cut-off values of 6.83 kPa, 6.98 kPa, 7.02 kPa, and 11.52 kPa for ≥F1, ≥F2, ≥F3, and F4, respectively. Recently, the diagnostic accuracy of the pSWE-Elast-PQ (EPQ) technique was assessed in a prospective, multinational cohort of 353 (suspected) NAFLD patients using VCTE as a reference standard [71]. The authors reported a strong correlation (Pearson R = 0.87; *p* < 0.0001) and concordance (Lin’s concordance correlation coefficient = 0.792) of EPQ with VCTE, and they concluded that the EPQ can reliably exclude NAFLD fibrosis < 6.0 kPa (<1.41 m/s) (AUC 0.94, Se 86%, Sp 89%). Optimal Youden’s index-derived cut-offs for stages of fibrosis were defined at ≥6.5 kPa (≥1.47 m/s) (AUC 0.94, Se 80%, Sp 95%) for significant fibrosis (≥F2), at ≥6.9 kPa (≥1.52 m/s) (AUC 0.949, Se 88, Sp 89%) for advanced fibrosis (≥F3), and at >10.4 kPa (>1.86 m/s) (AUC 0.949, Se 87%, Sp 94% for cirrhosis (F4). For ruling out and ruling in ≥F3, the authors established the following cut-offs: <7 kPa (AUC 0.949. Se 86%, Sp 90%), and ≥10.9 kPa (AUC 0.949, Se 60%, Sp 99%), respectively. This cut-offs should be further evaluated also by using the liver biopsy as a reference standard. Future studies using this technique would be of interest. 

### 2.3. Two-Dimensional Shear Wave Elastography (2D-SWE)

The use of two-dimensional shear wave elastography (2D-SWE) for the non-invasive assessment of liver fibrosis has grown rapidly. For LSM, 2D-SWE produces dynamic stress in multiple focal zones using the same ARFI technique, resulting in a quantitative elastography measurement, which is also expressed in kPa or m/s. Briefly, the radiation force is induced in tissues by focused ultrasonic beams at a very high frame rate sequence capable of catching, in real time, the transient propagation of the resulting shear waves [73] (Figure 3). This imaging modality is performed using a conventional ultrasound probe during a standard intercostal ultrasonographic examination. The Doppler-like acquisition estimates the share wave speed over a region of interest (ROI). A colored ROI map is created and is superimposed on the B-mode image, providing stiffness information [74]. The Young’s modulus (E) is determined by the equation E = 3pc2, where *p* is the tissue density (constant), and c is the shear wave speed. Values between 4.5 and 5.5 kPa using the SuperSonic (SSI) equipment were reported in the healthy population; at present, SSI is the most valid 2D-SWE technique [26]. 

Liver stiffness of less than or equal to 5 kPa (1.3 m/sec) has a high probability of being normal [28]. Five measurements would be considered to be adequate if a quality assessment was provided by the manufacturer [58]. Currently, the guidelines recommend a minimum of three measurements, and the results should be expressed as the median together with the IQR [26]. The variability between consecutive liver stiffness acquisitions, assessed by means of the interquartile range-to-median ratio, is the most important quality criterion; when this ratio is higher than 30% for measurements given in kilopascals or higher than 15% for measurements given in meters per second, the accuracy of the technique is reduced [28]. 

Narrow intercostal spaces, a high body mass index (BMI), and a thoracic wall thickness of above 25 mm were associated with a higher rate of invalid measurements [75]. The main confounders for elevated liver stiffness are the same as those for the other techniques. Necroinflammation, congestion, mechanical cholestasis, food intake, and alcohol consumption can influence the LS results. Other diseases that can independently increase the liver stiffness are amyloidosis, lymphomas, and extramedullary hematopoiesis [56]. 

#### 2.3.1. Performance of 2D-SWE for Liver Fibrosis Assessment in NAFLD and NASH 

Nowadays, almost all manufacturers have fitted their newest ultrasound scanners with liver stiffness (LS) modules. We briefly present the recent published results for 2D-SWE-SSI, Hologic Supersonic Imagine, Aix-en-Provence, France, Canon Medical Systems, formerly Toshiba Medical Systems, Otawara, Japan, and General Electric, GE Healthcare, Arlington Heights, IL, USA. It is of importance to mention that the elastography techniques of different manufacturers do not share the same distribution and range of values. These discrepancies constitute a major issue for the clinical practice, as ideally, the same rule-in and rule-out criteria, including for advanced fibrosis, should be applicable to all 2D-SWE techniques. Nevertheless, the variability between the liver stiffness measurements among different systems has decreased due to the continuous efforts of the Quantitative Imaging Biomarkers Alliance (QIBA) [28]. 

A recently published study compared the performances of 2D-SWE from the newest apparatus GE’s (2D-SWE-GE—Logiq E10), Canon’s (2D-SWE-Canon–Aplio i800) Supersonic Imagine (2D SWE-SSI—Mach30 Aixplorer), and VCTE (FibroScan with M and XL probe for BMI > 30 kg/m^2^) for measuring liver stiffness [76]. For the included patients (for which 1437 alcohol/NAFLD was the main etiology, > 66%), the same operator measured the liver stiffness with 2D-SWE-SSI (Supersonic Image) plus one of the following devices: 2D-SWE-GE (n = 314), 2D-SWE-Canon (n = 311), and VCTE-M probe (n = 812). The authors concluded that VCTE-M and 2D-SWE-SSI (Mach30 Aixplorer) values shared the highest correlation and concordance coefficients (0.933 and 0.920, respectively), and all four techniques had similarly low values for excluding advanced chronic liver diseases (ACLD) in larger populations, but discrepancies were observed in the high percentile values. 

Table 1 summarizes some of the diagnostic cut-off values and performances of 2D-SWE for diagnosing fibrosis in biopsy-proven NAFLD. We believe it would be of great interest for all 2D-SWE-equipped US machines to meet agreement between cut-off values that could rule out and rule in advanced fibrosis in NAFLD and NASH. 

The 2D-SWE performance used to identify patients with compensated advanced chronic liver disease (cACLD) according to the Baveno VI criteria based on VCTE cut-off values was assessed in 2020 in a cohort of 1219 consecutive patients (345 characterized as NAFLD) with chronic liver diseases who underwent both VCTE and 2D-SWE examinations on the same day [92]. According to the authors, 2D-SWE (performed using Aixplorer™ ultrasound device—Supersonic Imagine, Aix-en-Provence, France) accurately identified the patients with cACLD according to the Baveno VI criteria based on the VCTE cut-off values (Pearson’s correlation coefficient, 0.882; *p* < 0.0001; Lin concordance coefficient, 0.846; *p* < 0.0001). A 10 kPa (Se 92%, Sp 87%) 2D-SWE cut-off value was used to rule out cACLD. 

#### 2.3.2. Evaluation of Hepatic Steatosis in NAFLD and NASH Using 2D-SWE Techniques 

ATI (attenuation imaging) is a recently developed technique (Aplio i800, Canon Medical Systems) that quantifies the US attenuation in the tissue within a large sample measurement and using real-time color mapping with the purpose of evaluating hepatic steatosis in a non-invasive manner [93,94]. With this technique, the attenuation coefficient can be calculated in decibels per centimeter per megahertz (dB/cm/MHz). 

For the examination to be adequately performed, the patient should fast for at least 6 h. The measurements should be obtained from the right lobe of the liver, through the intercostal spaces, and with the patient lying in the supine position, as described for 2D-SWE. Reliable measurements are usually defined as the median value of five measurements performed in a homogeneous area of liver parenchyma, with IQR/M < 30% and R^2^ > 0.90 (R^2^—quality parameter recommended by the manufacturer) [95]. In a recent published study, 132 patients with biopsy-proven NAFLD underwent an ATI evaluation. The AUC values for ATI in diagnosing different grades of steatosis were 0.98 (Se 85%, Sp 97%), 0.94 (Se 95%, Sp 80%), and 0.94 (Se 100%, Sp 83%) for diagnosing ≥ S1 (≥5% steatosis), ≥S2 (≥34% steatosis), and ≥S3 (≥67% steatosis) steatosis grades, respectively. The authors concluded that ATI is a highly feasible method for the quantification of liver steatosis. For patients with ascites, shear wave can be freely generated with focused ultrasound through the water, contrary to the mechanical vibration of CAP. 

Table 2 summarizes some of the diagnostic cut-off values and performances of ATI for detecting different grades of liver steatosis in biopsy-proven NAFLD. 

Table 2 summarizes some of the diagnostic cut-off values and performances of UGAP for detecting different grades of liver steatosis in biopsy-proven NAFLD.

#### 2.3.3. Evaluation of Hepatic Inflammation in NAFLD and NASH by 2D-SWE Using Dispersion Slope (DS) 

Shear wave dispersion imaging is a newly developed ultrasound incorporated technology that analyzes the frequency dispersion properties of a tissue. It provides new information regarding the tissue viscosity, with recently described applicability in assessing diffuse liver diseases [102]. Preliminary data on its clinical utilization, characterization of chronic liver diseases, and detection of allograft damage after liver transplantation are now available [103,104]. 

Table 3 summarizes some of the preliminary data on the performance of DS for detecting lobular inflammatory activity in biopsy-proven NAFLD. 

## 3. US Elastography-Based Scores to Detect or Predict NASH 

### 3.1. Agile 3+ and Agile 4 Scores for Identification of Advanced Fibrosis and Cirrhosis in Suspected NAFLD 

Two FibroScan-based scores, Agile 4 and Agile 3+, have been recently developed and validated for identifying cirrhosis (F4) or advanced fibrosis (≥F3), respectively, in patients with NAFLD in the setting of specialized liver clinics [106]. Each score combined the LSM, AST/ALT ratio, platelets, sex, and diabetes status, as well as age, for Agile 3+. For Agile 4, two cut-offs were established, 0.251 and 0.565, to rule out and rule in F4, respectively, (rule out cut-off, which achieved sensitivity of ≥85%, and rule in cut-off, which achieved specificity of ≥95% for the diagnosis of cirrhosis). For Agile 3+, two cut-offs were established, 0.451 and 0.679, to rule out and rule in ≥F3, respectively, (rule out cut-off, which achieved sensitivity of ≥85%, and rule in cut-off, which achieved specificity of ≥90% for the diagnosis of advanced fibrosis). The authors concluded that by combining these simple clinical parameters together with routine laboratory biomarkers and LSM by VCTE, it is possible to reduce the number of cases with indeterminate results when one is performing VCTE only. As they are designed for secondary and tertiary care settings, these scores could be used to predict the clinical outcomes and monitor the disease progression. 

### 3.2. The Fibroscan-AST (FAST) Score for Identification of At-Risk NASH 

The FibroScan-AST (FAST) (CAP, LS, and AST) score for the non-invasive identification of patients with non-alcoholic steatohepatitis with significant activity and fibrosis was developed and validated in 2020 [107]. The score is aimed to non-invasively identify the “at-risk NASH” patients, characterized by the presence of NASH, an elevated NAFLD activity score (NAS ≥ 4), and significant fibrosis (F ≥ 2). Two cut-off values were proposed to rule out (≤0.35) and rule in (≥0.67) at-risk NASH, with an AUC of 0.95 (Se 0.90%, Sp 0.53%) and an AUC of 0.95 (Se 48%, Sp 90%), respectively. Table 4 summarizes some of the diagnostic cut-off values and the FAST score performance for diagnosing at-risk NASH in biopsy-proven NAFLD. The score should serve as a non-invasive method for identifying patients with NASH that could benefit from clinical trials or treatments when they become available. Applied to the general population, a score that stratifies the patients into low-risk and high-risk NASH patients could also reduce the number of referrals to specialists and decrease the number of unnecessary biopsies in patients with a slight possibility of having significant fibrosis [108]. 

### 3.3. Multiparametric Risk Scores for Identification of NASH Using Ultrasound Elastography

Multiparametric elastography scores combining more than two variables are now presenting great interest in the field of NAFLD and NASH. As recently reported, by using a simple scoring system of two US parameters, ATI (attenuation coefficient—for steatosis assessment) and DS (dispersion slope—for lobular inflammatory activity assessment), namely the unweighted sum of each score, the authors showed good correlation with the NAFLD activity score on histopathologic examination, with a correlation coefficient of 0.84 (95% CI: 0.77, 0.88; *p* < 0.001) [87]. The LAD-NASH score (liver stiffness, attenuation coefficient, and dispersion slope) could identify at risk NASH with AUC of 0.86 (Se 90.7%, Sp 61.4%) and 0.88 (Se 100%, Sp 33.7%) in the derivation and validation cohorts, respectively [113]. In the future, these scores could be used to identify, in a non-invasive manner, NASH patients that could benefit from inclusion in clinical trials and from pharmacologic therapy. 

## 4. Clinically Significant Portal Hypertension (CSPH) in Advanced NASH and NASH-Related Cirrhosis

Clinically significant portal hypertension (CSPH), the main driver of cirrhosis decompensation, is defined by a hepatic venous pressure gradient (HVPG) ≥ 10 mmHg [114]. Mild or subclinical portal hypertension is a hemodynamic abnormality defined by an HVPG between 6–9 mmHg [115]. An HVPG value ≥ 10 mmHg identifies patients with compensated cirrhosis at elevated risk of decompensation, this being of outmost importance for disease management and prognosis. In patients with CSPH, decompensation can be prevented by using non-selective beta blockers (NSBBs), preferably carvedilol [116]. The measurement of HVPG is hampered by its invasive nature, so in recent years, the scientific community concentrated its efforts in finding non-invasive and more available markers that could predict the presence of CSPH. In this respect, measuring LS by VCTE proved its utility LS ≥ 10 kPa was considered suggestive for cACLD, and ≥15 kPa highly suggestive for cACLD [117,118,119,120]. 

Based on the ANTICIPATE model, patients with LS by VCTE values between 20–25 kPa and platelet count < 150 × 10^9^/L or LS values between 15–20 kPa and platelet count < 110 × 10^9^/L have a CSPH risk of at least 60% [121]. The results from the ANTICIPATE study also contributed to a proposed non-invasive risk stratification algorithm (Baveno VII) with reliable cut-off values for ruling-out or ruling-in the presence of CSPH in patients with cACLD. According to the guidelines, LS by VCTE ≤ 15 kPa plus platelet count ≥ 150 × 10^9^/L rules-out CSPH (sensitivity and negative predictive value > 90%) in patients with cACLD. In patients with virus- and/or alcohol-related cACLD and non-obese (BMI < 30 kg/m^2^) NASH-related cACLD, a LS value by VCTE of ≥25 kPa is sufficient to rule-in CSPH (specificity and positive predictive value > 90%), defining the group of patients at risk of endoscopic signs of portal hypertension and at higher risk of decompensation [51]. In further attempts to validate this criteria, the ANTICIPATE model did not perform as well in patients with obese NASH. This led to the newly proposed ANTICIPATE-NASH model that incorporated the BMI alongside LS and PLT for a better prediction of CSPH in this specific subgroup [121]. The authors concluded that in patients with cACLD portal hypertension is present in more than 90% when the etiology is alcohol liver disease (ALD), hepatitis C virus (HCV), and hepatitis B virus (HBV), whereas in patients with NASH and especially in those with obesity, the prevalence of portal hypertension was much lower. The LSM ≥ 25 kPa cut-off was not specific enough to rule-in CSPH in obese patients with NASH, but LSM ≤ 15 kPa plus platelet count ≥ 150 × 10^9^/L could rule-out CSPH in most etiologies of chronic liver disease. 

By applying the actual Baveno VII criteria to rule-in or rule-out CSPH, a substantial proportion of patients (approx. 40%) remain unclassified in the “grey zone”, thus reducing the number of patients who may benefit from NSBB treatment without undergoing invasive evaluation such as HVPG measurement. Jachs et al. recently proposed (monocentric, retrospective study) a new stratification algorithm based on LS, PLT and the von Willebrand antigen to PLT ratio (VITRO) for refined diagnosis of CSPH. In their prospective cohort consisting of 302 cACLD patients (NASH etiology 13.2%), CSPH prevalence was 62.3% in the derivation cohort, while 45.7% were ‘unclassified’ according to Baveno VII criteria. By sequential application of the Baveno VII criteria and the newly proposed VITRO score, the number of previously “unclassified” patients diminished by almost 70%. A VITRO score ≤ 1.5 and ≥ 2.5 could rule-out (sensitivity, 97.7%; negative predictive value, 97.5%) and rule-in (specificity, 94.7%; positive predictive value, 91.2%), respectively, CSPH for the patients previously “unclassified” by applying the Baveno VII criteria [122]. 

Spleen Stiffness Measurement (SSM) estimates portal hypertension in patients with chronic liver disease (CLD). According to the guidelines, two spleen stiffness cut-off values (<21 kPa and>50 kPa) may, respectively, be applied for the rule-out and rule-in of CSPH in compensated advanced chronic liver disease (cACLD) due to viral hepatitis [123]. 

In this respect, another group proposed the add-on of the SSM for reducing the number of patients unclassified in the “grey zone” after applying the Baveno VII criteria [124]. The Combined Baveno VII—SSM model required, for ruling-out the presence of CSPH, meeting at least 2 of the 3 following criteria: LSM ≤ 15 kPa, PLT ≥ 150 × 10^9^/L, SSM ≤ 40 kPa. For ruling in the presence of CSPH meeting at least 2 of the 3 following criteria: LSM ≥ 25 kPa, PLT < 150 × 10^9^/L, SSM > 40 kPa. The authors concluded that the addition of SSM (40 kPa) to the model significantly reduced the gray zone to 7%–15%, maintaining adequate negative and positive predictive values (NPV and PPV, respectively ≥ 90%). 

As recently stated, patients with advanced NAFLD had a higher prevalence of portal hypertension–related decompensation events at any value of HVPG as compared with advanced HCV patients. Additionally, decompensation in advanced NAFLD may occur at lower HVPG levels than in patients with other etiologies [125]. Defining “the right” values for the NASH patients, especially for those with cACLD or experiencing CSPH is particularly important as for NASH compensated cirrhosis, the risk of progression to decompensation is 10% while subsequent liver related events or death among NASH adults with first decompensation reach 30% per year [7]. How this knowledge could be adapted for NASH patients is still ongoing research. 

## 5. A Multistep Approach to Non-Invasively Identify NAFLD-Related Fibrosis 

Given the high prevalence and incidence of NAFLD worldwide, screening for advanced fibrosis (≥F3) should start at the level of primary healthcare settings, and US elastography plays a significant role in the risk stratification and management of these patients [123,126]. 

For identifying NAFLD-related fibrosis, we propose a simple algorithm that implies clinical and laboratory data, the applicability of FIB-4, and LSM by VCTE as the most accepted and validated non-invasive elastography technique for liver fibrosis assessment. As highlighted in Figure 4 and according to the present guidelines [26,28,51,123], the elastography cut-off values that could rule-out the presence of significant fibrosis in NAFLD and that could describe a liver with a high probability of being normal by SWE and ARFI, respectively, were also included. A similar algorithm that is used for the stratification of patients with “high-risk” NAFLD using non-invasive tests, such as FIB-4, LSM by VCTE, or an ELF score, has been recently proposed by Younossi et al. [127]. 

### Screening for NAFLD-Related Fibrosis and Algorithm Development 

The American Gastroenterological Association (AGA) Clinical Pathway outlined the characteristics of patients at risk of NAFLD-related fibrosis (Figure 4) [128], among them were the patients with ≥2 metabolic risk factors [129]. 

The Fibrosis-4 Index (FIB4) and NFS (NAFLD fibrosis score) are two non-patented scores that could rule out or rule in advanced fibrosis (≥F3) in NAFLD. From the proposed cut-offs, one with a high sensitivity (1.3 for FIB-4, and -1.455 for NFS) and another one with a high specificity (3.25 for FIB-4 and 0.676 for NFS) are mentioned [123]. 

FIB4 (PLT—platelets; AST—aspartate aminotransferase; ALT—alanine aminotransferase) was primarily validated as an accurate marker of fibrosis in patients with hepatitis C viral (HCV) infections [131], with its applicability being extended over time in multiple clinical settings. An FIB4 < 1.3 could reliably exclude ≥F3 in patients with NAFLD, with a negative predictive value (NPV) of ≥90% [132,133,134]. An FIB4 cut-off of < 2.0 to rule out ≥F3 in patients aged ≥65 years led to a drop in the gray-zone results, making referrals to hepatologists more sustainable [135]. FIB4 ≥ 2.67 was strongly associated with ≥F3, all-cause mortality, and liver-related adverse outcomes among NAFLD patients [136,137], while values of ≥3.25 could rule in ≥F3, with the patients in the grey zone (1.30–3.25) requiring further investigations [138]. 

By using VCTE, patients could be stratified into three fibrosis risk groups: <8.0 kPa (low risk of ≥F3), 8.0–12.0 kPa (intermediate risk of ≥F3), and >12.0 kPa (high risk of ≥F3), while ≥ 20 kPa could positively diagnose cirrhosis without the need for a liver biopsy, with Sp 95% [130,139]. As stated in the current guidelines, FIB-4 < 1.30 and LSM by VCTE < 8 kPa rule out the presence of ≥F3 [123], while VCTE ≥ 12–15 kPa was recommended to rule in ≥F3 after considering the causes of false positives [139]. When one is assessing the prognosis, patients with FIB 4 <1.30 and those with FIB 4 ≥ 1.30 and further VCTE < 8.0 kPa had excellent prognoses, while those with FIB 4 ≥ 1.30 and VCTE ≥ 8.0 kPa presented an increased risk of liver-related events LREs [140]. 

If the VCTE is not available, NFS < −1.455 or patented tests such as Enhanced Liver Fibrosis Test (ELF^TM^) < 9.8, FibroMeter^TM^ < 0.45, or FibroTest^®^ < 0.48 could be used to rule out ≥F3 in patients with NAFLD, if they are available [123]. 

Over the years, NITs were suggested to be less accurate in patients with T2DM [141,142], and FIB4 did not qualify as the best NIT for first-line liver fibrosis assessment of this subgroup of patients [143]. In this context, VCTE could play an important role for primary risk stratification. 

A new two-step algorithm based on first-line VCTE or FibroMeter^TM^ (FM) fibrosis assessments, then second-line combined FibroMeter^TM^VCTE for further stratification was recently proposed for T2DM [143]. This algorithm maintained low rates of false negatives (FN) and false positives (FP) in this subgroup. The proposed cut-offs for ruling out ≥F3 were VCTE < 8.0 kPa or FM < 0.26 in the first-line assessment. For the second step, patients with VCTE ≥ 8.0 kPa or FM ≥ 0.26 with further FibroMeter^TM^-VCTE > 0.69 were stratified as being at risk for ≥F3. For FibroMeter^TM^ VCTE < 0.32, the presence of ≥F3 was ruled out. 

The American Gastroenterological Association (AGA) recently proposed a new three-step approach (launched and examined 2322 participants with risk factors for NAFLD-related fibrosis) to better identify the patients at risk of presenting significant fibrosis (≥F2) in a cost-effective manner. The pathway was based on identifying the population presenting risk factors for NAFLD, performing FIB4 and considering the presence of diabetes, and performing LSM by VCTE, respectively. According to the authors, a patient classified by FIB4 as having an indeterminate risk of ≥F2 between 1.3–2.67 (or 2–2.67 if age 65+) should further undergo an assessment for the presence of diabetes (DM). If diabetes is absent, the patient could be considered at low risk of having ≥F2, with a follow-up evaluation recommended in 2–3 years. If diabetes is present, the patient is then scheduled to undergo LSM by VCTE [128]. 

As presented above, T2DM and NAFLD are particularly challenging subsets of patients, as they tend to present more advanced stages of fibrosis at a younger age, but also a lower accuracy of NITs when one is assessing the risk of fibrosis. However, over referral should not be a solution for this type of patients, and we believe that personalized, non-invasive diagnostic algorithms should be assessed and further validated for this specific subgroup.

## 6. Future Perspectives 

In patients with NAFLD, a further validation of the biomarkers that could predict liver-related outcomes, identify patients who may benefit from treatment, and predict a response to therapeutic interventions would accelerate drug development [144] and provide a more accurate and cost-efficient disease management strategy. We believe that new updates to the existing ultrasound technology and new techniques, such as multiparametric ultrasound, should only improve the management of NAFLD patients. The assessment of inflammation and steatosis by multiparametric US, together with fibrosis, if it is accurate and reproducible, could identify the patients at risk of NASH and bring new improvements to risk stratification. 

Recently, optimal LS by VCTE thresholds for predicting the progression to cirrhosis among patients with bridging fibrosis (F3) and development of liver-related events among patients with cirrhosis were published (≥16.6 kPa and ≥30.7 kPa, respectively) [145]. We believe that further efforts will be invested in this direction, and multiparametric ultrasound could play an important role. 

NAFLD has also been viewed as an independent correlate of cardiovascular risk (CVR), however, the correlation between hepatic fibrosis and CVR is rather unclear [146]. As liver biopsy still represents the gold standard for fibrosis assessment, promising noninvasive tests and techniques have been studied for CVR assessments in NAFLD [147], including hepatic elastography [148]. We believe that future steps will move towards longitudinal studies, with appropriate follow-ups evaluating the association between NAFLD, NASH, and significant cardiovascular events. Considering the present and future opportunities to perform multiparametric ultrasound (with the combined assessments of fibrosis, steatosis, and inflammation), we believe that hepatic elastography could be included in CVR stratification strategies.

The discovery of new, reliable, non-invasive biomarkers based on metabolomics, proteomics [149], or circulating extracellular vesicles with a rich cargo of bioactive molecules may serve as reliable non-invasive “liquid biopsies” for NASH diagnosis and the assessment of the disease severity [150], and together with ultrasound elastography, a combined “multiomic” strategy could increase the accuracy of non-invasive diagnosis algorithms for NAFLD and maybe provide highly specific, sensitive, and accurate new diagnostic algorithms. 

## 7. Conclusions 

With this review, we provide the comprehensive, up-to-date research on the applicability of ultrasound elastography in adult patients with non-alcoholic fatty liver disease (NAFLD) and propose a multistep approach towards identifying NAFLD-related fibrosis. 

Identifying advanced fibrosis in NAFLD and non-alcoholic steatohepatitis (NASH) is of outmost importance for risk stratification and disease management. Non-invasive tests (NITs) and diagnostic algorithms are now available for this purpose, and they are adapted to both primary care settings and more specialized and advanced health services. Cut-offs have important implications for the sensitivity, the specificity, and the size of the indeterminate results when ruling in or ruling out a certain condition, including advanced fibrosis. The cut-offs applied in practice are usually heterogeneous between different healthcare settings and among different ultrasound machines, so we should consider the benefits of using standardized cut-offs. Vibration-controlled transient elastography (VCTE) is still the most utilized and validated elastography method for liver fibrosis assessments. When one is interpreting the VCTE results, the error factors should always be considered. New multiparametric ultrasound techniques are emerging, with the potential to quantify fibrosis, steatosis, and probably, inflammation, presenting promising results, even for non-alcoholic steatohepatitis (NASH). Considering that VCTE is not widely available and the results are influenced by different factors, such as hepatic inflammation, we have high expectations from the new multiparametric ultrasound techniques as possible game changers in NASH diagnosis and management.

## Figures and Tables

**Figure 1 diagnostics-13-00788-f001:**
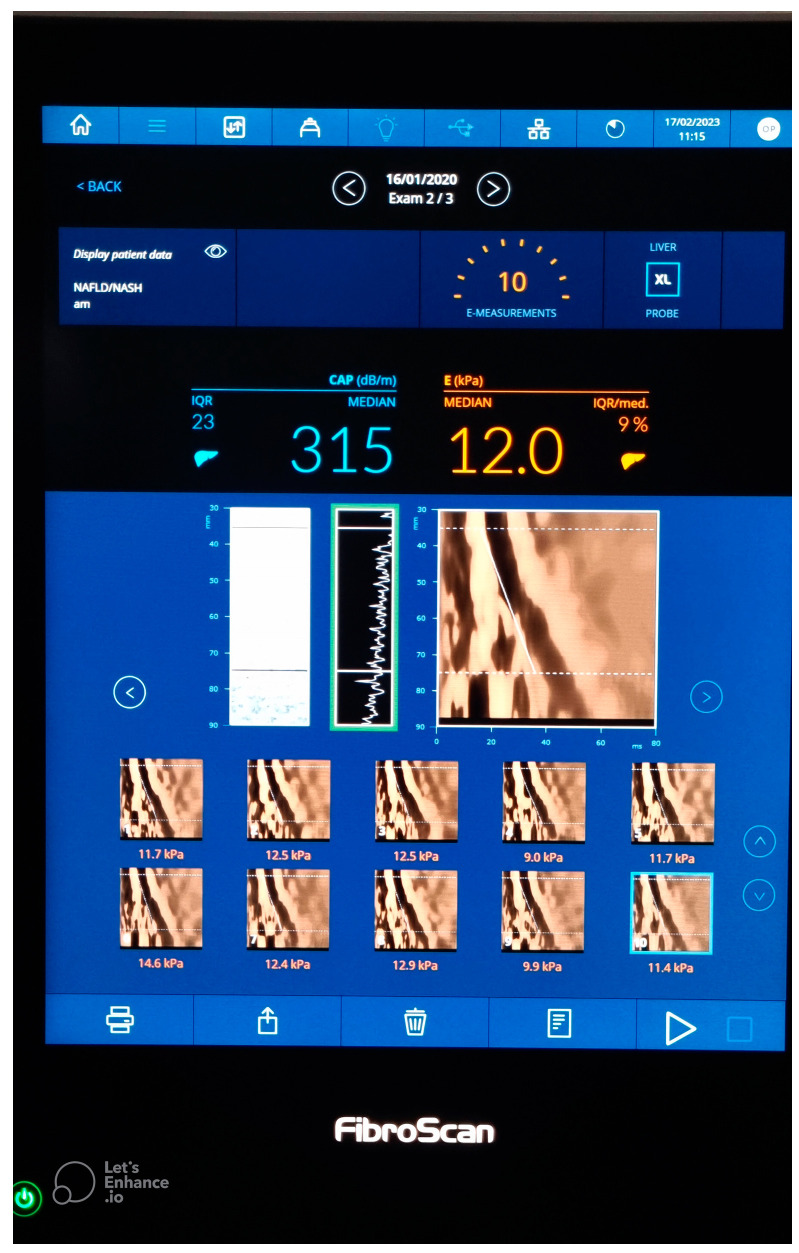
Vibration- controlled transient elastography (VCTE).

**Figure 2 diagnostics-13-00788-f002:**
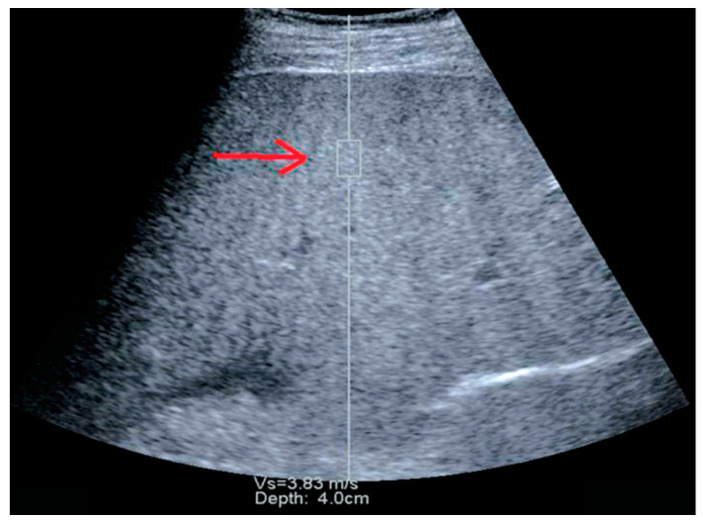
Point shear wave elastography (pSWE) using Siemens equipment. The “red arrow” is pointing to the region of interest (ROI).

**Figure 3 diagnostics-13-00788-f003:**
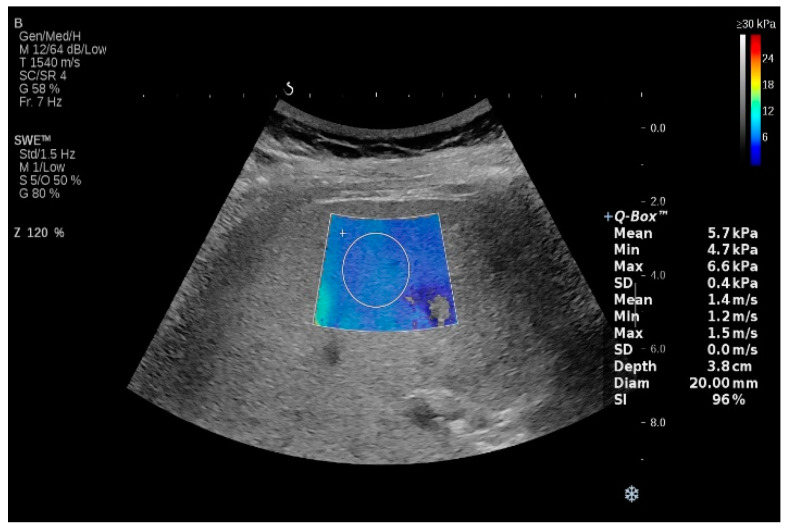
Two-dimensional shear wave elastography (2D-SWE) using SuperSonic equipment.

**Figure 4 diagnostics-13-00788-f004:**
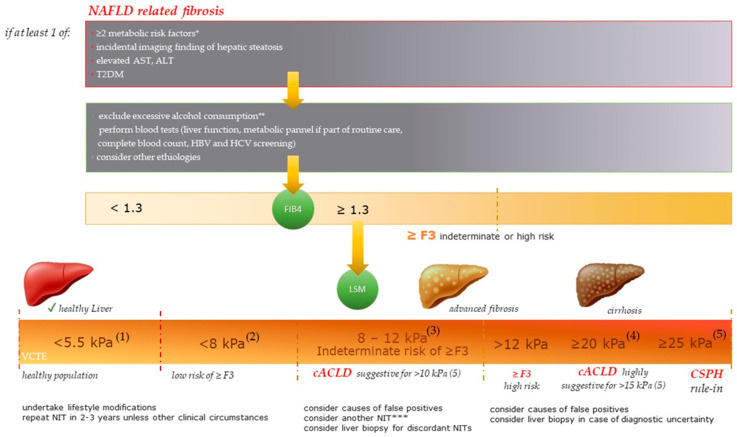
Screening algorithm for NAFLD-related Fibrosis. * the metabolic conditions include central obesity—waist circumference with ethnicity-specific cut-offs, serum triglycerides ≥150 mg/dL or specific treatment for hypertriglyceridemia, reduced serum high-density lipoprotein cholesterol <40 mg/dL in males (<50 mg/dL in females) or those undergoing a specific treatment, systolic blood pressure ≥130 mmHg, or diastolic blood pressure ≥85 mmHg or specific treatment, raised fasting plasma glucose between 100 mg/dL and 125 mg/dL (prediabetes) [128,129]; T2DM—type two diabetes mellitus; AST—aspartate aminotransferase; ALT—alanine aminotransferase; ** ≥30 g/day for men and ≥20 g/day for women; HBV—hepatitis B virus; HCV—hepatitis C virus; FIB4—Fibrosis-4 Index; LSM—liver stiffness measurement; T2DM—type two diabetes mellitus; VCTE—vibration-controlled transient elastography, whereby the values obtained with the XL probe are usually lower than those with the M probe [26]. (1) shear wave elastography (SWE) within the normal range can rule out significant liver fibrosis when in it is in agreement with the clinical and laboratory background [26], an LS ≤ 5 kPa presents high probability of being normal as recommended in the “rule of four” for acoustic radiation force impulse (ARFI) techniques [28], (2) VCTE < 8 kPa rules out advanced fibrosis in NAFLD [123], (3) the 8 kPa and 12 kPa dual cut-offs have a better diagnostic accuracy of NAFLD-related cACLD than the previously established cut-offs do [115] (cACLD is a term introduced in 2015 to describe the spectrum of advanced fibrosis (≥F3) and cirrhosis (F4) in compensated patients) (4) cut-off result from a recently published meta-analysis [130], and (5) according to the Baveno VII criteria [51], *** ELF^TM^ < 9.8 or FibroMeter^TM^ < 0.45 or FibroTest^®^ < 0.48 to rule-out ≥F3 in NAFLD [123]. CSPH—clinical significant portal hypertension.

**Table 1 diagnostics-13-00788-t001:** Performance of LS cut-off values by 2D-SWE for detecting different stages of liver fibrosis in biopsy-proven NAFLD.

Fibrosis Stage			≥F1			≥F2			≥F3			≥F4	
Study	Manufacturer	Cut-Off (kPa)	AUROC	Se/Sp (%)	Cut-Off (kPa)	AUROC	Se/Sp (%)	Cut-Off (kPa)	AUROC	Se/Sp (%)	Cut-Off (kPa)	AUROC	Se/Sp (%)
Cassinotto [77], 2016 (n = 291)	SuperSonic Imagine	N/S	N/S	N/S	6.3	0.86	90/50	8.3	0.89	91/71	10.5	0.88	97/72
		-	-	-	8.7	0.86	71/90	10.7	0.89	71/90	14.5	0.88	58/90
Lee [78], 2017 (n = 94)	SuperSonic Imagine	N/S	N/S	N/S	8.3	0.759	87/55.3	10.7	0.809	90/61.2	15.1	0.906	90/78
Takeuchi [79], 2018 (n = 71)	SuperSonic Imagine	6.61	0.82	79/67	11.57	0.75	52/44	13.07	0.82	63/57	15.7	0.9	100/82
Herrmann [80], 2018 * (n = 156)	SuperSonic Imagine	N/S	N/S	N/S	7.1	0.855	93.8/52	9.2	0.928	93.1/80.9	13	0.917	75.3/87.8
Ozturk [81], 2020 (n = 116)	SuperSonic Imagine	N/S	N/S	N/S	8.4	0.73	77/66	9.3	0.82	84/70	N/S	N/S	N/S
Sharpton [82], 2021 (n = 114)	SuperSonic Imagine	7.5	0.72	53.4/90.2	7.7	0.84	75.7/85.7	7.7	0.88	90/77.7	9.3	0.93	88.9/84.8
Selvaraj [25], 2021 * (n = 488)	SuperSonic Imagine	N/S	N/S	N/S	8.3–11.6	0.75	71/67	9.3–13.1	0.72	72/72	14.4–15.7	0.88	78/84
Zhou [83], 2022 (n = 116)	SuperSonic Imagine	N/S	N/S	N/S	10	0.86	72.9/80.4	11.6	0.89	80/88.4	12.6	0.9	76.5/92.2
		-	-	-	7.3	0.86	95.8/41.3	8.3	0.89	92/43.5	8.6	0.9	94.1/46.7
		-	-	-	11.2	0.86	58.8/95.7	12.8	0.89	68/94.2	13.5	0.9	70.6/93.5
Cassinotto [84], 2021 (n = 443)	Hologic, Marlborough	N/S	N/S	N/S	N/S	0.84	N/S	9.4	0.88	N/S	N/S	0.86	N/S
		-	-	-	-	-	-	8	0.88	Se > 90	-	-	-
		-	-	-	-	-	-	10.5	0.88	Sp > 90	-	-	-
Lee [85], 2021 (n = 102)	Canon Medical Systems	6.3	0.82	63/88	7.6	0.87	89/77	9	0.95	100/85	N/S	N/S	N/S
		4.9	0.82	91/52	6.3	0.87	95/59	9	0.95	100/85	-	-	-
		7	0.82	56/92	10.2	0.87	58/94	10.2	0.95	89/91	-	-	-
Sugimoto [86], 2020 (n = 111)	Canon Medical Systems	1.33 ***	0.79	64/90	1.4 ***	0.88	75/86	1.4 ***	0.9	85/79	1.55 ***	0.95	100/82
Jang [87], 2022 (n = 132)	Canon Medical Systems	6.4	0.89	73/92	7	0.92	97/84	7	0.91	100/77	8	0.93	100/78
Kuroda [88], 2021 (n = 202)	GE Healthcare	6.43	0.815	83.2/65.9	7.25	0.87	87.4/68.9	8.4	0.91	87.5/75.9	10.04	0.933	88.9/82.6
Zhang [89], 2022 (n = 100)	GE Healthcare	1.27 ***	0.65	91.2/11.6	1.49 ***	0.81	90.5/43	1.46 ***	0.85	93.8/39.3	1.59 ***	0.91	100/61.7
		1.75 ***	0.65	33.3/90.7	1.79 ***	0.81	47.6/91.1	1.78 ***	0.85	62.5/90.5	1.81 ***	0.91	83.3/90.4
Ochi [90], 2012 training set (n = 106)	Hitachi Medical Systems	2.47 ^	0.838	64.9/96.9	2.67 ^	0.853	86/88.6	3.02 ^	0.878	88.2/91.5	3.36 ^	0.965	100/85.6
validation set (n = 75)		2.47 ^	N/S	75/88.4	2.67 ^	N/S	92.3/89.8	3.02 ^	N/S	88.9/96.5	3.36 ^	N/S	100/95.3
Xiao [91], 2017 ** (n = 429)	N/S	N/S	N/S	N/S	2.67–9.4	0.89	85/94.4	3.02–10.6	0.91	89.9/91.8	3.36	0.97	100/85.6

* meta-analyses; ** meta-analysis including pediatric population; *** results expressed in m/s; ^—hepatic elastic ratio measured; AUROC—area under the ROC curve; kPa—kilopascals; n—population; Se—sensitivity; Sp—specificity; N/S—not specified.

**Table 2 diagnostics-13-00788-t002:** Performance of ATI/UGAP cut-off values by 2D-SWE for detecting different grades of liver steatosis in biopsy-proven NAFLD.

Fibrosis Stage			≥S1 (5%)			≥S2 (>33%)			≥S3 (>66%)	
Study	Manufacturer	Cut-Off (dB/cm/MHz)	AUROC	Se/Sp (%)	Cut-Off (dB/cm/MHz)	AUROC	Se/Sp (%)	Cut-Off (dB/cm/MHz)	AUROC	Se/Sp (%)
Bae [96], 2019 (n = 108)	Canon Medical Systems	0.635	0.843	74.5/77.4	0.7	0.886	86.4/81.4	0.745	0.926	100/82.4
Tada [97], 2019 biopsy-proven steatosis (n = 148)		0.66	0.85	67.8/87.6	0.67	0.91	92/83.7	0.68	0.91	100/75.2
biopsy-proven NAFLD (n = 38)		0.71	0.77	61.5/91.7	0.74	0.88	76.9/88	0.74	0.86	100/71.4
Dioguardi [98], 2020 (n = 101)		0.69	0.805	76/86	0.72	0.892	96/74	N/S	N/S	N/S
		>0.9	0.805	19/100	-	-	-	N/S	N/S	N/S
		<0.44	0.805	100/0	-	-	-	N/S	N/S	N/S
Sugimoto [86], 2020 (n = 111)		0.67	0.88	75/100	0.72	0.86	90/66	0.86	0.79	61/85
Lee [85], 2021 (n = 102)		0.64	0.93	75/95	0.7	0.88	84/76	0.73	0.82	86/69
		0.6	0.93	90/73	0.64	0.88	93/66	0.68	0.82	93/55
		0.63	0.93	78/91	0.77	0.88	54/91	0.82	0.82	50/91
Kuroda [99], 2021 (n = 105)		0.64	0.876	81.9/75	0.71	0.883	88.9/67.6	0.75	0.908	88.5/80.8
Jang [87], 2022 (n = 132)		0.62	0.94	85/97	0.7	0.94	95/80	0.78	0.94	100/83
Fujiwara [100], 2018 (n = 163)	GE Healthcare	0.53	0.9	81.2/87.1	0.6	0.95	85.7/81.5	0.65	0.959	80.4/90
Kuroda [88], 2021 (n = 202)		0.493	0.891 *	79.1/90	0.654	0.909 *	84.2/86.6	0.691	0.924 *	88.2/83.3
Ogino [101], 2021 (n = 84)		0.6	0.94	86.7/88.9	0.71	0.95	85.7/91.8	0.72	0.88	85.7/80

* Compared with CAP; AUROC—area under the ROC curve; Se—sensitivity; Sp—specificity; N/S—not specified; n—population. The ultrasound-guided attenuation parameter (UGAP) embedded by GE Healthcare can also be used for hepatic steatosis assessments. UGAP measures the attenuation coefficient (AC) (dB/cm/MHz) of the B-mode ultrasonic signal via general ultrasonography [100]. The measurements should be taken from the right liver lobe using an intercostal approach during short breath holds, with the patient in the fasting state for at least 4 h and in a supine position, as mentioned for 2D-SWE [99]. In a recent published study, 84 patients with biopsy-proven NAFLD underwent UGAP evaluations [101]. UGAP presented a good diagnostic performance of AC values for steatosis scores of S1, S2, and ≥S3 in the ROC curve analysis (0.94, 0.95, and 0.88, respectively). The authors concluded that AC values obtained using UGAP could be a useful new method for quantifying steatosis in NAFLD.

**Table 3 diagnostics-13-00788-t003:** Performance of DS cut-off values by 2D-SWE for detecting lobular inflammatory activity in biopsy-proven NAFLD.

Fibrosis Stage			≥I1			≥I2			≥I3	
Study	Manufacturer	Cut-Off (m/s/kHz)	AUROC	Se/Sp (%)	Cut-Off (m/s/kHz)	AUROC	Se/Sp (%)	Cut-Off (m/s/kHz)	AUROC	Se/Sp (%)
Sugimoto [86], 2020 (n = 111)	Canon Medical Systems	8.5	0.95	94/100	9.9	0.81	89/66	12.5	0.85	83
Lee [85], 2021 (n = 102)	Canon Medical Systems	10.5	0.89	74/96	10.5	0.85	85/74	11.7	0.78	100/63
		9.1	0.89	91/58	9.9	0.85	92/62	11.7	0.78	100/63
		10.4	0.89	75/92	12.8	0.85	50/90	19.7	0.78	0/100
Jang [87], 2022 (n = 132)	Canon Medical Systems	10.8	0.86	82/82	82/82	0.86	90/77	11.6	0.79	100/61

AUROC—area under the ROC curve; Se—sensitivity; Sp—specificity; N/S—not specified; n—population; I1—mild lobular inflammation grade; I2—moderate lobular inflammation grade; I3—severe lobular inflammation grade. In conclusion, due to the recent advancements of 2D-SWE techniques and promising future developments, we can obtain, in a non-invasive manner, valuable information regarding fibrosis, steatosis, and possible inflammation in both chronic and acute liver diseases [104,105].

**Table 4 diagnostics-13-00788-t004:** Performance of FAST core cut-off values for detecting at-risk NASH (NASH + NAS ≥ 4 + F ≥ 2) in biopsy-proven NAFLD.

	Rule out at-Risk NASH				Grey Zone	Rule in at-Risk NASH		
Study	Cut-Off	AUROC	Se/Sp (%)	NPV	Interval	Cut-Off	Se/Sp (%)	PPV
Newsome [107], 2020 Derivation (n = 350)	≤0.35	0.8	90/53	0.85	0.35–0.67	≥0.67	48/90	0.83
External Validation (n = 1026)	≤0.35	0.85	89/64	0.94	0.35–0.67	≥0.67	49/92	0.69
Woreta [109], 2022 (n = 585)	≤0.35	0.81	91/50	0.9	0.35–0.67	≥0.67	52/87	0.69
Lee [110], 2022 (n = 251)	≤0.35	0.714	93.1/35.2	0.926	0.35–0.67	≥0.67	56.9/77.1	0.5
De [111], 2022 biopsy-proven NAFLD (n = 60)	≤0.35	0.81	91/14	0.88	0.35–0.67	≥0.67	73/67	0.33
biopsy-proven NASH (n = 17)	≤ 0.55	0.81	90/45	0.95	0.55–0.78	≥ 0.78	64/94	0.7
Cardoso [112], 2022 (n = 287)	≤0.35	0.78	78.8/64	0.878	0.35–0.67	≥0.67	41.2/89.1	0.618

AUROC—area under the ROC curve; Se—sensitivity; Sp—specificity; NPV—negative predictive value; PPV—positive predictive value; N/S—not specified; n—population.

## Data Availability

Not applicable.

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
