# Peer review of "How to Identify Advanced Fibrosis in Adult Patients with Non-Alcoholic Fatty Liver Disease (NAFLD) and Non-Alcoholic Steatohepatitis (NASH) Using Ultrasound Elastography—A Review of the Literature and Proposed Multistep Approach"

_diagnostics, 2023, doi:10.3390/diagnostics13040788_

Round 1
Reviewer 1 Report
The author described developments in the noninvasive assessment for the early detection of NAFLD. The review is good and brings all the information that we need about the Elastography. Most of the references are new.
1. Will be good to include some others figures to illustrate better the review.
Author Response
Response for Reviewer 1
Dear Reviewer,
We really appreciate and thank you very much for your constructive feedback. Please find below our response to your suggestions.
Suggestion 1: The author described developments in the noninvasive assessment for the early detection of NAFLD. The review is good and brings all the information that we need about the Elastography. Most of the references are new. Will be good to include some others figures to illustrate better the review.
Response 1: We thank you for bringing into light this important aspect regarding the visual impact of our proposed article. We added in the updated form of the manuscript new images presenting some of the most important elastography techniques that are described in the article.
Reviewer 2 Report
The present paper is a comprehensive state-of the art review of the literature concerning the non-invasive evaluation of NAFLD. The authors do a great job of describing each method in detail: equipment, work method including confounding factors and cut-offs. Every statement is supported by an extensive reference list. They also identify points for future research.
I have 2 suggestions:
1. To include the new nomenclature for NAFLD, metabolic associated fatty liver disease:
A new definition for metabolic dysfunction-associated fatty liver disease: An international expert consensus statement. Eslam, Mohammed et al. Journal of Hepatology, Volume 73, Issue 1, 202 - 209
2. since we are facing an obesity epidemic in children, maybe include pediatric data
Author Response
Response for Reviewer 2
Dear Reviewer,
We really appreciate and thank you very much for your constructive feedback. Please find below our response to your suggestions.
Suggestion 1: The present paper is a comprehensive state-of the art review of the literature concerning the non-invasive evaluation of NAFLD. The authors do a great job of describing each method in detail: equipment, work method including confounding factors and cut-offs. Every statement is supported by an extensive reference list. They also identify points for future research. To include the new nomenclature for NAFLD, metabolic associated fatty liver disease: A new definition for metabolic dysfunction-associated fatty liver disease: An international expert consensus statement. Eslam, Mohammed et al. Journal of Hepatology, Volume 73, Issue 1, 202 – 209.
Response 1: We thank you for this especially important comment. We added in “Introduction” a paragraph concerning the newly proposed definition of NAFLD – metabolic associated fatty liver disease (MAFLD).
Suggestion 2: Since we are facing an obesity epidemic in children, maybe include pediatric data.
Response 2: We thank you for this suggestion. To specify more clearly the topic of our review, we updated the title from “How to Identify Advanced Fibrosis in Patients with Non-alcoholic Fatty Liver Disease (NAFLD) and Non-alcoholic Steatohepatitis (NASH) Using Ultrasound Elastography. A Multistep Approach” to “How to Identify Advanced Fibrosis in Adult Patients with Non-alcoholic Fatty Liver Disease (NAFLD) and Non-alcoholic Steatohepatitis (NASH) Using Ultrasound Elastography. A Review of the Literature and Proposed Multistep Approach”.
Indeed, the obesity pandemic in children is of great concern and of great interest among the research community. Considering this important impact, an impressive amount of very important research works has been published in the last years and we definitely believe that a state-of the art review on this topic could detail and put together all the important aspects of the topic. As clinical practitioners that manage adult patients in daily practice, we did not want to minimize any important aspect regarding the applicability of ultrasound elastography in children, as the amount of recently published data on this topic is quite representative and, in our opinion, this population presents its own particularities. In this respect, we believed that targeting in our review only the adult population would be more accurate.
Reviewer 3 Report
GENERAL COMMENT
The Authors performed an interesting and very informative review exploring the non invasive prediction of advanced NAFLD by ultrasound elastography. Some comments may be raised at improving the quality of the manuscript.
SPECIFIC COMMENTS
- NAFLD has been viewed as an independent cardiovascular risk factor. Non-invasive assessment of liver fibrosis by non-invasive testing (serum liver biomarkers as well as sonoelastography) has been correlated with cardiovascular risk in NAFLD patients (e.g. Ballestri S, et al. Diagnostics. 2021; 11(1):98.; Ballestri S et al. Metab Target Organ Damage 2023;3:1.)
Please comment and update literature.
- Literature should be updated with more recent studies:
Younossi Z, Alkhouri N, Cusi K, Isaacs S, Kanwal F, Noureddin M, Loomba R, Ravendhran N, Lam B, Nader K, Racila A, Nader F, Henry L. A practical use of noninvasive tests in clinical practice to identify high-risk patients with nonalcoholic steatohepatitis. Aliment Pharmacol Ther. 2023 Feb;57(3):304-312. doi: 10.1111/apt.17346. Epub 2022 Dec 13. PMID: 36511349.
Loomba R, Huang DQ, Sanyal AJ, Anstee QM, Trauner M, Lawitz EJ, Ding D, Ma L, Jia C, Billin A, Huss RS, Chung C, Goodman Z, Wong VW, Okanoue T, Romero-Gómez M, Abdelmalek MF, Muir A, Afdhal N, Bosch J, Harrison S, Younossi ZM, Myers RP. Liver stiffness thresholds to predict disease progression and clinical outcomes in bridging fibrosis and cirrhosis. Gut. 2023 Mar;72(3):581-589. doi: 10.1136/gutjnl-2022-327777. Epub 2022 Sep 9. PMID: 36750244; PMCID: PMC9905707.
Author Response
Response to Reviewer 3
Dear Reviewer,
We really appreciate and thank you very much for your constructive feedback. Please find below our response to your suggestions.
Suggestion 1: The Authors performed an interesting and very informative review exploring the non invasive prediction of advanced NAFLD by ultrasound elastography. Some comments may be raised at improving the quality of the manuscript. NAFLD has been viewed as an independent cardiovascular risk factor. Non-invasive assessment of liver fibrosis by non-invasive testing (serum liver biomarkers as well as sonoelastography) has been correlated with cardiovascular risk in NAFLD patients (e.g. Ballestri S, et al. Diagnostics. 2021; 11(1):98.; Ballestri S et al. Metab Target Organ Damage 2023;3:1.)
Response 1: We thank you for this comment. We updated the manuscript with information regarding the applicability and our belief for future perspectives regarding the role of noninvasive tests in discriminating cardiovascular risk in the setting of NAFLD in the section entitled “6. Future perspectives”.
Suggestion 2: Literature should be updated with more recent studies:
Younossi Z, Alkhouri N, Cusi K, Isaacs S, Kanwal F, Noureddin M, Loomba R, Ravendhran N, Lam B, Nader K, Racila A, Nader F, Henry L. A practical use of noninvasive tests in clinical practice to identify high-risk patients with nonalcoholic steatohepatitis. Aliment Pharmacol Ther. 2023 Feb;57(3):304-312. doi: 10.1111/apt.17346. Epub 2022 Dec 13. PMID: 36511349.
Loomba R, Huang DQ, Sanyal AJ, Anstee QM, Trauner M, Lawitz EJ, Ding D, Ma L, Jia C, Billin A, Huss RS, Chung C, Goodman Z, Wong VW, Okanoue T, Romero-Gómez M, Abdelmalek MF, Muir A, Afdhal N, Bosch J, Harrison S, Younossi ZM, Myers RP. Liver stiffness thresholds to predict disease progression and clinical outcomes in bridging fibrosis and cirrhosis. Gut. 2023 Mar;72(3):581-589. doi: 10.1136/gutjnl-2022-327777. Epub 2022 Sep 9. PMID: 36750244; PMCID: PMC9905707.
Response 2: We appreciate this comment. We updated the reference list and included in the revised version of the manuscript the relevant information derived from this articles.
Reviewer 4 Report
The review proposed by Taru et al provides a comprehensive picture of the available ultrasound elastography methods used for fibrosis assessment in NAFLD and NASH. The methods are well described with rather new references and a good reflection of the data literature.
I have a concern and I am not sure about what the authors suggested by “a multistep approach”. Mentioning this in the title makes you think that you may find a sequence of methods to help you determine more easily advanced fibrosis. Instead, the review does not include a direct approach but several described methods. This must be clarified within the title and text.
Also, my recommendation is that in such a manuscript consider is necessary to elaborate another chapter in which to share the author’s opinion on future perspectives for the setting they discussed.
Author Response
Response to Reviewer 4
Dear Reviewer,
We really appreciate and thank you very much for your constructive feedback. Please find below our response to your suggestions.
Suggestion 1: The review proposed by Taru et al provides a comprehensive picture of the available ultrasound elastography methods used for fibrosis assessment in NAFLD and NASH. The methods are well described with rather new references and a good reflection of the data literature. I have a concern and I am not sure about what the authors suggested by “a multistep approach”. Mentioning this in the title makes you think that you may find a sequence of methods to help you determine more easily advanced fibrosis. Instead, the review does not include a direct approach but several described methods. This must be clarified within the title and text.
Response 1: We thank you for this especially important feedback. We reformulated the section regarding the “multistep approach” and moved it to a newly introduced section entitled “5. A Multistep Approach to Noninvasively Identify NAFLD Related Fibrosis”. Out “multistep” proposed algorithm is detailed in Figure 4.
We updated the title from “How to Identify Advanced Fibrosis in Patients with Non-alcoholic Fatty Liver Disease (NAFLD) and Non-alcoholic Steatohepatitis (NASH) Using Ultrasound Elastography. A Multistep Approach” to “How to Identify Advanced Fibrosis in Adult Patients with Non-alcoholic Fatty Liver Disease (NAFLD) and Non-alcoholic Steatohepatitis (NASH) Using Ultrasound Elastography. A Review of the Literature and Proposed Multistep Approach”.
Suggestion 2: Also, my recommendation is that in such a manuscript consider is necessary to elaborate another chapter in which to share the author’s opinion on future perspectives for the setting they discussed.
Response 2: We very much appreciate this suggestion. We added in the manuscript a new section entitled “6. Future perspectives”.
Round 2
Reviewer 3 Report
The manuscript has substantially immproved.
Reviewer 4 Report
I appreciate taking into account my recommendations!